# Local heterogeneity in Lassa fever serology in rural Nigeria: Implications for vaccine trial site selection

David Simons[1]*, Christina Harden[1], Natalie Imirzian[2], Katharine E. T. Thompson[1], Nzube Michael Ifebueme[3], Sunday Eziechina[4], Helen Ignatius[3], Diana Marcus[3], Fisayomi Aderibigbe[3], James T. Koninga[3], Martin Meremikwu[3], Lina Moses[5], David W. Redding[2], Sagan Friant[1]

1 Department of Anthropology, Pennsylvania State University, State College, Pennsylvania, United States of America, 2 Science Department, Natural History Museum, London, United Kingdom, 3 Calabar Institute for Tropical Disease Research and Prevention, University of Calabar, Calabar, Cross River, Nigeria, 4 Nigeria Centre for Disease Control, Abuja, Nigeria, 5 Celia Scott Weatherhead School of Public Health and Tropical Medicine, Tulane University, New Orleans, Louisiana, United States of America

* dzs6259@psu.edu

## Abstract

### Introduction

Lassa virus (LASV) causes significant morbidity in West Africa, yet vaccine trial planning is hampered by a lack of high-resolution, community-level seroprevalence data. We aimed to characterize the fine-scale spatial and demographic heterogeneity of LASV exposure in rural Nigeria to inform site selection strategies.

### Methods

We conducted a cross-sectional serosurvey across nine villages in three Nigerian states (Benue, Ebonyi, and Cross River). We recruited 1,874 individuals and tested for LASV IgG using the highly specific Panadea LASV IgG ELISA. We employed Bayesian hierarchical models to estimate seroprevalence and investigated 21 pre-specified demographic, environmental, and behavioral risk factors for association with seropositivity. Village-specific transmission dynamics were explored using Bayesian generalized additive models for age-stratified seroprevalence.

### Results

The overall model-based IgG seroprevalence was 3.2% (95% CrI: 2.5–4.0%). We observed marked fine-scale heterogeneity, with village-level estimates ranging from 0.8% to 6.5%. Age-seroprevalence curves suggested? divergent transmission dynamics, ranging from cumulative endemic exposure to recent focal outbreaks in younger cohorts. Likely constrained by low overall prevalence and high exposure ubiquity, univariable analyses detected no strong or consistent associations between

**Data availability statement:** Zenodo provides the permanent Digital Object Identifier (DOI) for the specific release of the code and data used in this manuscript. As GitHub repositories are not permanent archives, retaining the Zenodo DOI is essential for long-term reproducibility and compliance with standard open science practices. Therefore, please typeset the statement as follows: "All de-identified data and associated R scripts required to reproduce these analyses are available on GitHub (https://github.com/RiskLabPSU/baseline-seroprev-public) and permanently archived on Zenodo (https://doi.org/10.5281/zenodo.19677372).

**Funding:** This work was supported by the joint National Science Foundation-National Institutes of Health-National Institutes of Food and Agriculture Ecology and Evolution of Infectious Disease Award (Grant #2208034 to SF) and the United Kingdom Research and Innovation Biotechnology and Biological Sciences Research Council (Grant BB/X005364/1 to DWR). DS, NI, KETT, NMI, HOI, DM, and JTK received salary support from these grants. The funders had no role in study design, data collection and analysis, decision to publish, or preparation of the manuscript.

**Competing interests:** The authors have declared that no competing interests exist.

seropositivity and the 21 pre-specified risk factors, including rodent consumption and agricultural practices. No significant village-wide spatial clustering of seropositive households was observed via Local Getis-Ord (Gi*).

## Discussion

LASV exposure in rural Nigeria is characterized by low overall prevalence punctuated by significant hyper-local variation. The lack of consistent individual-level risk factors and the divergent age-exposure profiles suggest that risk may be influenced by stochastic, localized ecological drivers or obscured by the temporal misalignment of cross-sectional surveys. Vaccine trial site selection must move beyond regional incidence data to incorporate interdisciplinary, One Health metrics including high-resolution human serosurveillance and longitudinal reservoir monitoring to identify active transmission hotspots.

## Author summary

**Why was this study done?** Lassa fever is a significant viral disease in West Africa, and several vaccine candidates are currently in development. To evaluate vaccine efficacy in Phase III clinical trials, researchers must identify communities with active, ongoing viral transmission. However, Lassa virus spillover is highly unpredictable, making it difficult to select trial sites based on regional maps or passive clinical reporting. We aimed to determine if current broad-scale risk models accurately reflect local exposure and to identify the specific drivers of infection at the community level.

**What did the researchers do and find?** We tested 1,874 individuals across nine rural villages in Nigeria for Lassa virus IgG antibodies. We found that infection rates were generally low (overall 3.2%) but exhibited significant fine-scale variation: seroprevalence ranged from less than 1% to over 6% between villages, even those in close proximity. We investigated 21 potential risk factors, including rodent consumption, agricultural practices, and household environment, but – likely constrained by the low overall infection rate - found no consistent associations with infection. Furthermore, the patterns of infection by age differed between villages, suggesting communities may experience differing transmission dynamics, ranging from stable, long-term exposure to more episodic transmission.

**What do these findings mean?** Our results show that Lassa virus risk is hyper-local and cannot be reliably predicted by broad regional maps or human behavior alone. This suggests that infection risk may be influenced by stochastic, local ecological factors, such as the fluctuating presence of the virus within specific rodent populations. For vaccine trials, this means that relying on regional incidence data may lead to selecting sites with insufficient transmission to

measure vaccine success. We suggest that trial planning must adopt a One Health design integrating active human serosurveillance with longitudinal monitoring of rodent reservoirs to accurately identify active transmission hotspots for successful vaccine testing.

## Introduction

Lassa virus (*Mammarenavirus lassaense*; Arenaviridae, LASV) is a zoonotic arenavirus endemic to West Africa, with regular outbreaks occurring in Nigeria, Guinea, Liberia, and Sierra Leone. It causes Lassa fever, a viral hemorrhagic illness responsible for an estimated 900,000 annual infections and over 200 deaths annually in Nigeria alone [1]. Clinical presentation ranges from asymptomatic infection to severe hemorrhagic fever with case fatality exceeding 20% in hospitalized patients, for whom treatment options remain limited [2].

The primary reservoir for LASV is *Mastomys natalensis*, a highly fecund, synanthropic rodent widely distributed across sub-Saharan Africa [3]. It thrives in agricultural and peri-domestic environments, creating substantial opportunities for human contact [4]. Infected rodents, particularly those infected vertically in utero, can become persistent shedders of the virus in excreta, while its fluctuating population dynamics influence seasonal transmission intensity [5,6].

Human infection occurs primarily through direct or indirect contact with infected rodents, their excreta, contaminated food, or aerosolized particles [7]. Occupational and domestic exposures, particularly in farming, food storage, and rodent hunting contexts, are commonly implicated. However, the relative contribution of these pathways remains debated, and seroprevalence surveys show that exposure is widespread yet highly variable, with most infections (~80%) being asymptomatic or subclinical, meaning case counts severely underestimate the true infection burden [8,9].

Environmental and ecological conditions strongly influence LASV spillover [10]. Land use changes like deforestation and agricultural intensification can increase human-rodent contact, while shifts in rodent community composition may also moderate risk [11–13]. Despite these broad-scale drivers, observed variation in LASV prevalence between ecologically similar areas highlights the critical importance of fine-scale factors [14]. These include social conditions, such as household structure, food storage methods, and specific human-animal interactions. Understanding this fine-scale variation is not solely academic but is a prerequisite for the successful deployment of future medical countermeasures

LASV is a designated priority pathogen for the World Health Organization and the Coalition for Epidemic Preparedness Innovations (CEPI), with multiple vaccine candidates currently advancing into clinical trials [15]. The success of future Phase III efficacy trials relies on identifying communities where incidence is sufficient to measure a vaccine effect. However, relying on broad-scale incidence data or passive surveillance to select these sites risks recruitment failure if transmission is more focal than regional risk maps suggest, resulting in prolonged timelines for case accrual sufficient to demonstrate vaccine efficacy [16].

Current Lassa fever surveillance relies primarily on passive case detection at sentinel hospitals, which overlooks mild infections and asymptomatic seroconversion in the community [17]. As a result, major gaps remain in our understanding of the null areas and regions that appear ecologically suitable but report few cases. To address this, we conducted a cross-sectional study in nine communities across three Nigerian states [18]. We combined serological testing with structured questionnaires to investigate fine-scale drivers of exposure. Our objectives were to: 1) estimate LASV seroprevalence at state and village levels; 2) characterize individual and household correlates of seropositivity; and 3) explore spatial heterogeneity in local transmission risk.

## Materials and methods

The primary objective of this study was to estimate the seroprevalence of LASV IgG in nine communities across three Nigerian states. Secondary objectives were to explore demographic, environmental, behavioral, and occupational correlates of seropositivity and to assess fine-scale spatial heterogeneity of exposure.

## Ethics statement

The study protocol was approved by the Institutional Review Board of the Pennsylvania State University, USA (STUDY00019989), and by the Nigerian National Health Research Ethics Committee (AEC/03/168/24). Additional approval was obtained from the health research ethics committees of the public health agencies in each participating state. The study was conducted in accordance with the Declaration of Helsinki. Enrolment involved engagement with community leaders. Written informed consent was obtained from all adults; assent and written parental consent were obtained for minors.

## Study design and participants

This cross-sectional baseline survey was conducted as part of the SCAPES longitudinal study [18]. Nine villages across three states (Benue [BN], Ebonyi [EB], and Cross River [CR]) were selected using a systematic framework designed to sample communities across a gradient of postulated drivers of LASV spillover [19]. These states were chosen to represent a range of environmental suitability as defined by national-scale mechanistic modelling, which categorized them into high, medium, and low-risk tiers [20].

Using the site selection tool [19], candidate settlements were identified via OpenStreetMap. Land cover characteristics (ESA WorldCover) were analyzed within a 2-km radius representing the home range of the reservoir species (*M. natalensis*) to ensure sites reflected rural agricultural areas. We specifically targeted medium-sized villages (100–500 households) and excluded sites with >60% forest cover or >20% built-up area. Final site selection was validated to ensure no systematic bias relative to other settlements in the region and to ensure the sampled villages appropriately reflected the local variation in cropland and grassland proportions. This approach was designed to represent a spectrum of LASV spillover risk across the three states, rather than targeting known hotspot communities, thereby providing a more representative baseline of regional seroprevalence.

To contextualize our findings within the national epidemiological landscape, we collated historical passive surveillance data from the Nigeria Centre for Disease Control (NCDC) [21]. Specifically, we extracted the number of years each corresponding Local Government Area (LGA) had reported confirmed Lassa fever cases prior to the survey [22].

Households were enumerated to estimate population size and plan recruitment. Villages ranged from 110 to 321 households (population 465–2,313; S1 Fig). To prevent demographic sampling bias (such as over-representing individuals who remain at home during the day), we employed a strict intra-household quota system. We aimed to recruit ~20% of households per village (target ~70), specifically enrolling exactly four individuals per household: one adult male, one adult female, one adolescent (12–18), and one randomly selected child (<12). This approach was designed to ensure the final cohort accurately reflected the balanced demographic structure of the communities. Children under 12 provided dried blood spot (DBS) samples but did not complete questionnaires.

Based on published LASV seroprevalence data [14,17,23], we assumed high-risk (>40% seroprevalence), medium-risk (10–20%), and low-risk (<5%) village categories. We estimated that 180 individuals would provide 80% power ($\alpha = 0.05$, $\beta = 0.2$) to detect significant differences in LASV seroprevalence between village categories. Household-level clustering of serostatus was expected; hence, the design was powered to detect differences at both village and household levels.

Systematic sampling was used to select households within each village. Routes were designed to approach every $n$th household (where $n$ is the total number of households divided by the target sample size). Three households (all in BN) declined participation; in these instances, the next household was enrolled. The participant flow diagram is available in S2 Fig.

## Data collection

A questionnaire covering demographic, environmental, behavioral, and occupational domains was administered [24]. DBS samples were collected by finger-prick on Whatman 903 cards, air-dried, and transported with desiccant. DBS was

selected as the primary sampling modality due to the logistical challenges of maintaining a continuous cold-chain for serum during extended sampling sessions (typically two weeks per village) in remote rural locations. Samples were eluted and tested for LASV IgG using the Panadea LASV ELISA kit following the manufacturer's protocol modified for DBS [25,26]. Specifically, two 6mm discs were extracted from the DBS samples of an individual and were eluted in a 300 μL solution of $1 \times PBS$ containing 0.2% liquid ammonium and 2.5 μL of a 25% $NH_3$ solution following previously validated protocols for LASV antibody recovery. Samples were incubated overnight at 4°C, and the resulting eluates were used directly for ELISA without further dilution. This recombinant nucleoprotein (NP)-based assay was selected for its high specificity compared to traditional whole-virus lysate assays, which are prone to cross-reactivity with other endemic arenaviruses. In analytical validation studies, this platform has demonstrated a diagnostic specificity of 100% and a sensitivity of 95% against reference cohorts [26]. Consistent with its use for identifying baseline exposure, the assay threshold was set according to the manufacturer's instructions for qualitative detection of IgG.

## Statistical methods

Analyses were performed in R (v4.2.3) utilizing the tidyverse package (v2) [27,28]. Visualizations were produced using the ggplot2 package (v4.0.01) and the gtsummary package (v2.4.0) for tables [29, 30]. Questionnaire responses were linked to individual and household records using unique identifiers. Household GPS coordinates were used for spatial analyses. Data cleaning involved range validation, logical consistency checks, and outlier flagging. Missingness was negligible (<1%); analyses were restricted to complete cases.

## Descriptive group comparisons

To compare characteristics between villages or states, we fitted separate Bayesian hierarchical models using the brms (v2.22.0) and RStan (v2.32.7) packages, specifically Gaussian linear models for continuous variables and multinomial logistic regression for categorical data [31,32]. Following the principles of Bayesian hierarchical modelling, we assessed group-level heterogeneity by examining the posterior distribution of the standard deviation (SD) of the random effect [33]. Unlike frequentist *p*-values or Odds Ratios (where the null is 1.0), the SD represents the magnitude of among-group variance on the link scale. We defined "strong evidence" of a notable difference when the 95% CrI of the SD was entirely displaced from zero, with a lower bound >0.1; this ensured the identified variation was of a meaningful magnitude rather than a statistical artefact of the regularizing prior [33].

## Estimating LASV seroprevalence

We estimated seroprevalence and 95% CrIs using Bayesian generalized linear mixed models (GLMM) with a binary outcome (seropositive/seronegative) and a single fixed effect for state or village. Posterior predictions generated marginal estimates averaged over sampled individuals. To explore age-related variation, we fitted a Bayesian generalized additive model (GAM) with village-specific smooth terms for age. Posterior expected probabilities were calculated across a grid of age values to obtain modelled estimates. The full posterior summaries (central estimates and CrIs) for these models are presented in S2 and S3 Tables. Differences classed as important are indicated in the main text and tables. We utilized weakly informative priors for all fixed effects (Normal(0, 5)) and random effects (Exponential(1)) to allow the data to drive inference while stabilizing computation in low-prevalence groups.

   We did not attempt to formally estimate the Force of Infection (FOI) using catalytic models. Such models typically assume a time-invariant infection hazard, which we deemed inappropriate given the evidence of stochastic, episodic transmission in this region. Furthermore, the low number of seropositive cases (N = 61) across a wide age range precluded the stable estimation of a time-varying FOI model, which would require substantially higher power to distinguish between age-dependent risk and historical fluctuations in viral circulation.

## Characterizing behavioral and exposure correlates of seropositivity

We investigated associations between LASV seropositivity and a range of *a priori* risk factors using a Bayesian GLMM framework, specifying a Bernoulli (logit) likelihood for serostatus as the binary outcome. Analyses were conducted at both individual and household levels, with explanatory variables grouped into four conceptual domains: demographic, environmental, behavioral, and occupational (S1 Table).

To account for potential non-independence of observations within villages, reflecting shared environmental, infrastructural, or cultural exposures, a random intercept for village was included in all models. Univariable models were fitted separately for each covariate to examine associations in isolation, adjusting only for village-level clustering. We evaluated the magnitude and uncertainty of potential risk factors by examining the posterior odds ratios (OR) and their 95% CrIs. An interval that included 1.0 was interpreted as showing no strong evidence for an association. Unlike the descriptive comparisons above, which used the random effect SD to identify village-level heterogeneity, these models report the fixed-effect OR to quantify the specific relationship between a covariate and the probability of seropositivity.

## Exploring spatial heterogeneity in LASV exposure

To assess spatial patterns in LASV exposure at the household level, we geocoded each sampled household and identified those with at least one seropositive individual. We used the terra (v1.8-60) and tidyterra (v1) packages to process spatial data [34,35]. We first applied Global Moran's I to test for overall spatial autocorrelation of household seropositivity within each village. To identify potential micro-scale clustering, we then used local Getis-Ord Gi* analysis using the spdep package (v1.3-13), which compares the value at each household (the proportion of seropositive members) to the values of neighboring households within a defined spatial lag (300m), generating a z-score indicating statistically significant hot spots (high-value clusters) and cold spots (low-value clusters) [36].

As a *post hoc* check on the relationship between hotspot classification and seropositivity, we fit a simplified logistic regression model. This model used household serostatus (presence/absence of seropositive individuals) as the outcome and compared two groups: households located in any Gi* defined cluster (Hotspot or Coldspot) versus those classified as Not Significant. This allowed us to robustly quantify the association between being in a statistically-defined cluster and the household's observed seropositivity.

Given the low seroprevalence and relatively small village area (median=6.7km$^2$), these analyses were intended as descriptive and exploratory, with a focus on identifying potential micro-scale heterogeneity rather than formally testing for clustering.

## Results

### Study population

Between 16 December 2023 and 22 July 2024, we enrolled 1,926 individuals (1,874 obtained DBS samples, 1,469 completed questionnaires) from 577 households across nine villages in three Nigerian states (BN, CR, EB). This sample represented 27% of households and 11% of the expected total village population (Table 1, Table 2 and S1 Fig).

### Households

Household composition and infrastructure showed notable variation across the study villages, particularly in household structure (Table 1). There was strong evidence of differences in household size, ranging from median sizes of 8 in Dyegh [BN] to 4 in Ofonekom [CR]. Similarly, the number of buildings per household and number of single-room buildings differed substantially between villages, with households in BN generally being larger and comprising more buildings.

Environmental characteristics were more consistent across sites. Proximity to bush (51%) and farms (66%), predominant sanitation method (field defecation: 80%), and rodent entry into homes (94%) did not show strong evidence of differing between villages.

Table 1. Household-level demographic and environmental characteristics.

| Characteristic | N | Overall N=577[1] | Benue [BN] | | | Cross River [CR] | | | Ebonyi [EB] | | |
|---|---|---|---|---|---|---|---|---|---|---|---|
| | | | Zugu N=63[1] | Dyegh N=64[1] | Ikyogbakpev N=64[1] | Okimbongha N=64[1] | Ogamanna N=66[1] | Ofonekom N=61[1] | Ezeakataka N=66[1] | Enyandulogu N=65[1] | Offianka N=64[1] |
| Number of households[2] | | | 205 (31%) | 305 (21%) | 149 (43%) | 191 (34%) | 231 (29%) | 110 (55%) | 321 (21%) | 281 (23%) | 381 (17%) |
| Household size (individuals)[3] | 577 | 6.0 (4.0) | 6.0 (4.0) | 8.0 (3.3) | 7.0 (4.0) | 6.0 (3.3) | 6.0 (3.8) | 4.0 (3.0) | 6.0 (3.8) | 6.0 (5.0) | 6.0 (3.0) |
| Household seropositivity[4] | 555 | | | | | | | | | | |
| All negative | | 496 (89%) | 58 (97%) | 58 (91%) | 58 (91%) | 43 (73%) | 53 (80%) | 42 (89%) | 64 (97%) | 61 (94%) | 59 (92%) |
| At least one positive | | 59 (11%) | 2 (3.3%) | 6 (9.4%) | 6 (9.4%) | 16 (27%) | 13 (20%) | 5 (11%) | 2 (3.0%) | 4 (6.2%) | 5 (7.8%) |
| Number of buildings[3] | 577 | 3.0 (2.0) | 4.0 (3.0) | 3.5 (2.0) | 4.0 (3.0) | 2.0 (0.0) | 2.0 (0.0) | 2.0 (0.0) | 3.0 (1.0) | 3.0 (2.0) | 3.0 (2.0) |
| Single-room buildings[3] | 577 | 1.0 (2.0) | 4.0 (3.0) | 3.0 (2.0) | 4.0 (3.0) | 1.0 (1.0) | 1.0 (1.0) | 1.0 (0.0) | 1.0 (1.0) | 2.0 (1.0) | 1.0 (1.0) |
| Proximity to bush[3] | 577 | 293 (51%) | 52 (83%) | 32 (50%) | 48 (75%) | 14 (22%) | 25 (38%) | 8 (13%) | 42 (64%) | 49 (75%) | 23 (36%) |
| Proximity to farm[3] | 577 | 382 (66%) | 59 (94%) | 49 (77%) | 58 (91%) | 12 (19%) | 14 (21%) | 10 (16%) | 59 (89%) | 58 (89%) | 63 (98%) |
| Type of toilet[3] | 577 | | | | | | | | | | |
| Plumbed toilet[3] | | 25 (4.3%) | 8 (13%) | 6 (9.4%) | 4 (6.3%) | 3 (4.7%) | 2 (3.0%) | 0 (0%) | 0 (0%) | 2 (3.1%) | 0 (0%) |
| Pit latrine[3] | | 77 (13%) | 16 (25%) | 13 (20%) | 12 (19%) | 5 (7.8%) | 3 (4.5%) | 0 (0%) | 8 (12%) | 4 (6.2%) | 16 (25%) |
| Open system[3] | | 13 (2.3%) | 0 (0%) | 0 (0%) | 0 (0%) | 0 (0%) | 0 (0%) | 1 (1.6%) | 8 (12%) | 3 (4.6%) | 1 (1.6%) |
| Field defecation[3] | | 460 (80%) | 37 (59%) | 45 (70%) | 48 (75%) | 56 (88%) | 61 (92%) | 60 (98%) | 50 (76%) | 56 (86%) | 47 (73%) |
| Other[3] | | 2 (0.3%) | 2 (3.2%) | 0 (0%) | 0 (0%) | 0 (0%) | 0 (0%) | 0 (0%) | 0 (0%) | 0 (0%) | 0 (0%) |
| Rodents enter home | 577 | 540 (94%) | 62 (98%) | 62 (97%) | 59 (92%) | 61 (95%) | 62 (94%) | 56 (92%) | 56 (85%) | 62 (95%) | 60 (94%) |

1 Median (IQR); n (%)

2 % refers to proportion of households enrolled from all households in the village

3 Indicates strong evidence of heterogeneity between villages for this variable. Posterior summaries quantifying these between-village differences are detailed in S4 Table

4 Maximum of 4 household members were tested irrespective of household size

**Table 2. Individual-level demographic, occupational, and behavioral characteristics.**

| Characteristic | N | Overall N=1,841[1] | Benue [BN] Zugu N=179[1] | Dyegh N=209[1] | Ikyog-bakpev N=208[1] | Cross River [CR] Okim-bongha N=225[1] | Ogamanna N=230[1] | Ofonekom N=143[1] | Ebonyi [EB] Ezeakataka N=234[1] | Enyan-dulogu N=208[1] | Offianka N=205[1] |
|---|---|---|---|---|---|---|---|---|---|---|---|
| Estimated village population | | | 1,227 (15%) | 930 (23%) | 2,242 (9%) | 1,196 (21%) | 1,401 (16%) | 464 (40%) | 2,125 (11%) | 1,790 (12%) | 2,310 (9%) |
| LASV Seropositive | 1,841 | 61 (3.3%) | 2 (1.1%) | 6 (2.9%) | 8 (3.8%) | 16 (7.1%) | 13 (5.7%) | 5 (3.5%) | 2 (0.9%) | 4 (1.9%) | 5 (2.4%) |
| Age (years) | 1,841 | 27 (32) | 30 (27) | 27 (31) | 27 (31) | 26 (28) | 25 (29) | 26 (25) | 28 (30) | 30 (34) | 26 (32) |
| Sex: Female | 1,841 | 948 (51%) | 83 (46%) | 104 (50%) | 95 (46%) | 121 (54%) | 114 (50%) | 75 (52%) | 127 (54%) | 113 (54%) | 116 (57%) |
| Education[2] | 1,402 | | | | | | | | | | |
| None | | 224 (16%) | 16 (12%) | 36 (23%) | 27 (17%) | 35 (21%) | 9 (5.1%) | 15 (13%) | 24 (14%) | 28 (18%) | 34 (22%) |
| Primary | | 535 (38%) | 53 (38%) | 51 (32%) | 51 (32%) | 42 (25%) | 61 (34%) | 25 (22%) | 110 (63%) | 80 (52%) | 62 (41%) |
| Secondary | | 581 (41%) | 64 (46%) | 64 (41%) | 81 (51%) | 80 (47%) | 97 (55%) | 67 (58%) | 41 (23%) | 43 (28%) | 44 (29%) |
| Post-secondary | | 62 (4.4%) | 6 (4.3%) | 6 (3.8%) | 1 (0.6%) | 13 (7.6%) | 10 (5.6%) | 9 (7.8%) | 1 (0.6%) | 4 (2.6%) | 12 (7.9%) |
| Born in study village[2] | 1,398 | 1,031 (74%) | 80 (58%) | 83 (53%) | 96 (61%) | 149 (88%) | 169 (95%) | 101 (87%) | 132 (75%) | 111 (72%) | 110 (72%) |
| Occupation: Farmer | 1,841 | 1,083 (59%) | 129 (72%) | 132 (63%) | 121 (58%) | 123 (55%) | 145 (63%) | 101 (71%) | 117 (50%) | 104 (50%) | 111 (54%) |
| Occupation: Agri-cultural worker | 1,841 | 151 (8.2%) | 8 (4.5%) | 9 (4.3%) | 9 (4.3%) | 13 (5.8%) | 21 (9.1%) | 3 (2.1%) | 31 (13%) | 32 (15%) | 25 (12%) |
| Occupation: Trader | 1,841 | 312 (17%) | 39 (22%) | 46 (22%) | 28 (13%) | 25 (11%) | 20 (8.7%) | 15 (10%) | 53 (23%) | 40 (19%) | 46 (22%) |
| Rodent consumption[2] | 1,404 | 777 (55%) | 127 (91%) | 144 (91%) | 144 (90%) | 113 (66%) | 130 (73%) | 72 (62%) | 14 (8.0%) | 7 (4.5%) | 26 (17%) |
| Past rodent con-sumption only[2] | 627 | 446 (71%) | 10 (83%) | 11 (79%) | 13 (81%) | 32 (56%) | 30 (63%) | 30 (68%) | 109 (67%) | 113 (76%) | 98 (78%) |
| Ever rodent consumption[2] | 1,404 | 1,223 (87%) | 137 (99%) | 155 (98%) | 157 (98%) | 145 (85%) | 160 (90%) | 102 (88%) | 123 (70%) | 120 (77%) | 124 (82%) |
| Cleaned rodent excreta | 1,407 | 1,231 (87%) | 95 (67%) | 143 (91%) | 130 (81%) | 157 (92%) | 167 (94%) | 110 (95%) | 155 (88%) | 133 (86%) | 141 (93%) |

1 n (%); Median (IQR)

2 Indicates strong evidence of heterogeneity between villages for this variable. Posterior summaries quantifying these between-village differences are detailed in S4 Table

Households employed various methods for rodent removal. In contrast to the uniformity of rodent entry, our analysis found strong evidence that the specific methods used, such as poison (73%), sticks (44%), and traps, varied substantially between villages. However, the ultimate outcome for captured rodents was largely consistent; 88% of households reported disposing of them. While disposal was uniform, the practice of consuming captured rodents did show notable variation between villages, consistent with the individual-level data (Table 2).

### Individuals

Demographics and behaviors varied notably across sites (Table 2). Migration patterns differed (Strong Evidence); the percentage of residents born in their current village was substantially lower in BN compared to CR. However, most participants were permanent residents (median 12 months/year). A balanced distribution by sex was achieved by the sampling design for all sites. Median age (Overall: 28, IQR: 31) was consistent across villages.

Educational attainment varied substantially; participants in BN and CR had higher rates of secondary education completion compared to EB. Subsistence farming was the most common occupation (59%), followed by trading (17%) and hired agricultural work (6%). Primary occupations did not show strong evidence of difference between villages.

Rodent consumption practices differed substantially. Current rodent consumption was common in most BN and CR villages (e.g., 91% in Dyegh [BN] and 73% in Ogamanna [CR]) but was rarer in the EB villages (e.g., 4.5% in Enyandulogu). Even among those not currently consuming rodents, the proportion who reported past consumption also varied, being generally higher in EB villages (e.g., 78% in Offianka) than elsewhere. Indirect rodent contact, such as cleaning rodent excreta, was nearly universal (99%) and did not differ between villages.

### LASV seroprevalence estimates

Of the 1,874 individuals tested, 61 were seropositive for LASV IgG, yielding a crude overall seroprevalence of 3.3%. These individuals belonged to 59 households, with only two households (both in Ikyogbakpev [BN]) containing more than one seropositive member. No seropositive individuals reported a prior diagnosis of Lassa fever.

Model-based estimates revealed marked heterogeneity in exposure risk across the study area (Table 3). The overall seroprevalence was 3.17% (95% CrI: 2.47-4.04%). This varied at the state level (CR = 5.13% [3.66–6.97%]; BN = 2.63% [1.56–4.06%]; EB = 1.65% [0.86–2.85%]) and showed even greater variation at the village level (ranging from Ezeakataka [EB] = 0.79% [0.13-2.45%] to Okimbongha [CR] = 6.48% [3.9-9.97%]) (Table 3.).

**Table 3. Model-based LASV IgG seroprevalence estimates at the state and village level.**

| State/Village | N | Seropositive | | 95% CrI | |
| --- | --- | --- | --- | --- | --- |
| | | n | % | Lower | Upper |
| **Benue** | 616 | 16 | 2.63 | 1.56 | 4.06 |
| Dyegh | 216 | 6 | 2.74 | 1.08 | 5.47 |
| Ikyogbakpev | 214 | 8 | 3.72 | 1.70 | 6.82 |
| Zugu | 186 | 2 | 1.03 | 0.17 | 3.28 |
| **Cross River** | 597 | 34 | 5.13 | 3.66 | 6.97 |
| Ofonekom | 140 | 5 | 2.67 | 0.96 | 5.73 |
| Ogamanna | 234 | 13 | 5.53 | 3.06 | 8.94 |
| Okimbongha | 223 | 16 | 6.48 | 3.90 | 9.97 |
| **Ebonyi** | 661 | 11 | 1.65 | 0.86 | 2.85 |
| Enyandulogu | 215 | 4 | 1.81 | 0.55 | 4.23 |
| Ezeakataka | 239 | 2 | 0.79 | 0.13 | 2.45 |
| Offianka | 207 | 5 | 2.31 | 0.83 | 4.88 |

When contextualized against historical national surveillance records, the sampled villages fell within three distinct Local Government Areas representing a gradient of historical case reporting: Izzi LGA in Ebonyi (cases reported in 7 historical years, most recently 2024), Ikom LGA in Cross River (2 years, most recently 2022), and Vande Ikya LGA in Benue (1 year, most recently 2022). Notably, marked heterogeneity in village-level seroprevalence was observed even among communities nested within the same historically active LGA, highlighting a disconnect between aggregate regional notifications and true community-level exposure.

The relationship between age and seropositivity was not uniform, revealing distinct, village-specific transmission dynamics (Fig 1). Two villages (Ogamanna [CR] and Ofonekom [CR]) exhibited the classic pattern of increasing seroprevalence with age, consistent with long-term, cumulative exposure. This trend was highly consistent in Ogamanna [CR] (Posterior probability of increase > 99%), while Ofonekom [CR] showed a similar but more uncertain trajectory (Probability ~76%). Notably, these two villages are geographically adjacent (< 5km apart), suggesting a more similar local ecology. Conversely, Okimbongha [CR] and Ezeakataka [EB] displayed profiles peaking in younger groups before declining, which may suggest more recent, focal exposure, or the preferential waning of older antibodies, though statistical evidence for the decline was weaker (Probability 60–66%). In remaining villages, seroprevalence remained consistently low across ages.

## Risk factors for seropositivity

Univariable models adjusting for village-level clustering found no strong evidence of association between seropositivity and most pre-specified risk factors (Fig 2). Odds ratios were generally close to 1.0 with wide intervals. Given these results

**Fig 1. Village-specific age-seroprevalence curves for Lassa virus.** Colored lines represent median posterior estimates from a Bayesian generalized additive model, and shaded ribbons show the 95% credible intervals. Villages are grouped by state (columns). Within each panel, background grey lines display the median estimates for the other two surveyed villages within the same state to provide local comparative context. Note that the y-axis scales vary between states to accommodate the marked regional differences in estimated seroprevalence.

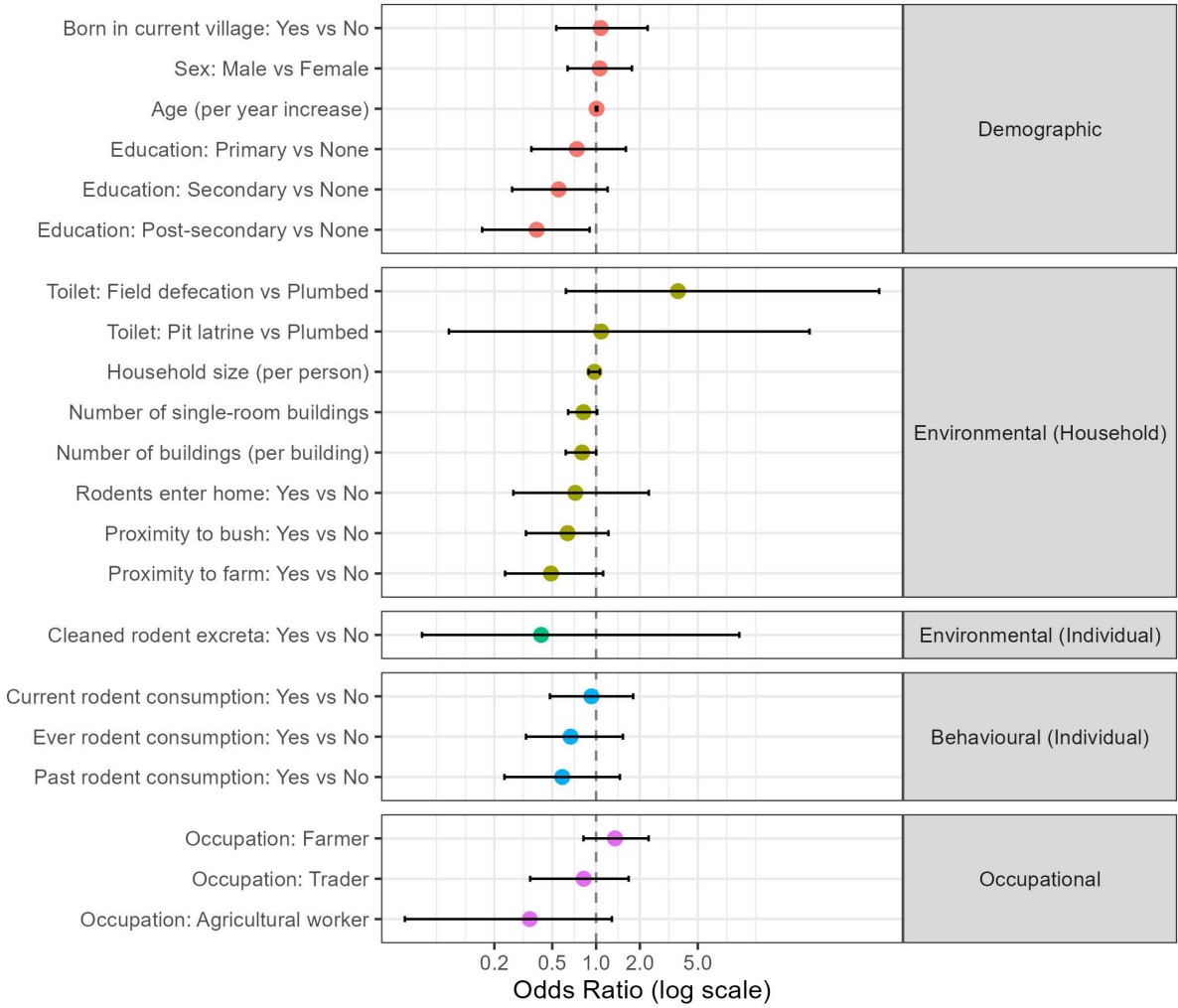

**Fig 2. Forest plot of univariable associations with Lassa virus seropositivity.** Points represent median posterior odds ratios derived from Bayesian hierarchical logistic regression models, with horizontal bars denoting 95% credible intervals. All models include a village-level random effect to account for spatial clustering. Variables are grouped by epidemiological domain. For continuous predictors (e.g., age, household size, number of buildings), the odds ratio reflects the change in odds per single-unit increase; for categorical predictors, the odds ratio reflects the comparison against a defined base-line reference group.

and the study's power limitations from overall low seropositivity, multivariable modelling was deemed unnecessary, we therefore only report univariable results below.

## Demographic correlates of LASV seropositivity

No meaningful association was found for age (OR: 1.01 [0.99–1.02]) or sex (OR: 1.06 [0.64–1.76]). Post-secondary edu-cation showed a protective trend (OR: 0.39 [0.17–0.90]), though likely confounded by age and socioeconomic status.

## Environmental and Behavioral correlates of LASV seropositivity

No clear associations were found for household-level variables such as household size (OR: 0.97, [0.89–1.06]), number of buildings (OR: 0.80, [0.62–1.00]), rodent entry into the home (OR: 0.72, [0.27–2.30]), or cleaning of rodent excreta

(OR: 0.42 [0.06–9.64]). Odds ratios for rodent consumption (OR: 0.93 [0.49–1.80]) and past rodent consumption only (OR: 0.59 [0.23–1.46]), and ever consuming rodents (OR: 0.67 [0.33-1.53]) were also uncertain.

Univariable analysis for all selected variables are shown in S1 Table.

## Spatial heterogeneity in LASV exposure

We assessed the spatial clustering of LASV seropositivity at the household level across nine villages. Global spatial autocorrelation, assessed using Moran's I, found no evidence of significant village-wide clustering in any site (all $p > 0.05$). Localized clustering was then evaluated using the Getis-Ord Gi* statistic, which identified a total of 27 hotspot and 6 cold-spot households across all villages (Fig 3). However, these statistical clusters did not align well with observed seropositivity; of the 59 seropositive households in the study, only 3.6% were located within a designated hotspot.

Logistic regression confirmed no significant difference in the odds of seropositivity based on cluster status ($p = 0.39$ OR $_{clustered\ vs.\ non-clustered}$ = 0.53 [0.12-2.3]). These results strongly indicate that, in our study sites, seropositive households were not concentrated in discernible spatial clusters.

## Discussion

This study found that LASV exposure in rural Nigeria was substantially lower than anticipated based on national risk models [10]. While the overall model-based IgG seroprevalence was 3.2% (2.5–4.0%), this aggregate figure masks marked heterogeneity at the local scale. Prevalence varied significantly between states (1.7% to 5.1%) and between villages (0.8% to 6.5%). These findings challenge broad-scale risk mapping, pointing to a transmission landscape that is generally low but punctuated by complex, hyper-local variation supported by three observations: divergent age patterns, an absence of risk factor correlation, and a lack of spatial clustering. While the degree of local variation reported here is notable, such fine-scale spatial heterogeneity is a well-documented feature of zoonoses, where transmission is highly dependent on micro-scale ecological interfaces [37].

Our seroprevalence estimates are comparable to those from some contemporary community-based studies in West Africa, though such comparisons are conditioned by differences in study populations and laboratory methods [14,38]. In Sierra Leone, seroprevalence was found to range between 0% and 88% at the village level across three districts, while in two Nigerian local government areas (Abuja Municipal Area Council and Ikorodu), seroprevalence was 33% and 18% respectively [14,39]. The high specificity of the commercial ELISA kit (LASV vs. non-LASV arenaviruses) used in this study may yield more conservative estimates than alternative serological assays, particularly given the known circulation of non-Lassa arenaviruses in Nigeria [26,40].

The relationship with age was more revealing. Two villages exhibited classic increasing seroprevalence with age, indicative of cumulative, endemic transmission. Conversely, two others showed peaks in younger groups, which may suggest dynamic, focal transmission [41]. Peaks in younger age groups could arise from recent increases in transmission or immunological factors like seroreversion. With antibody waning estimated at ~3% annually [42], the decline in older cohorts likely reflects the loss of detectable markers from historical infections. This implies that cross-sectional seroprevalence may serve only as a moving window of recent exposure (e.g., the last 15–20 years) rather than a cumulative record of lifetime risk. Consequently, the distinct peak in younger age groups observed in villages like Okimbongha [CR] could signal more recent transmission events concentrated in younger cohorts, while the low prevalence in older adults reflects a mixture of lower recent exposure and the decay of historical antibodies. From a trial planning perspective, these communities with steep age-seroprevalence gradients in younger cohorts likely represent the most viable candidates for efficacy studies, as they may indicate active, ongoing viral circulation rather than solely historical exposure.

While our univariable models did not identify age as a significant population-wide correlate of seropositivity (OR: 1.01 [0.99–1.02]), the village-specific GAMs reveal that age-related risk is not uniform across the study area. This suggests that

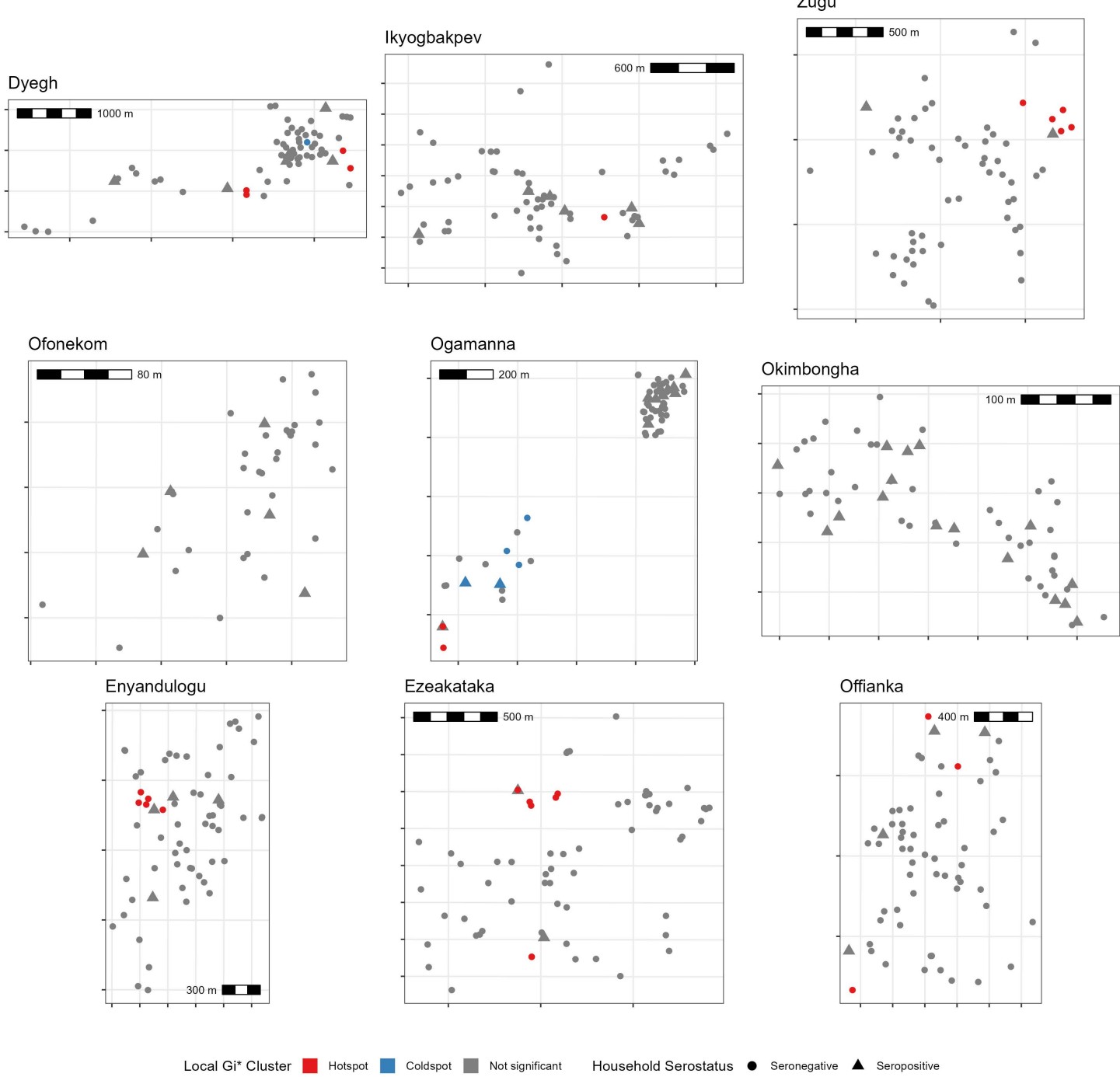

**Fig 3. Spatial distribution of household LASV seropositivity and Local Gi\* clustering.** Each panel represents a single village, showing jittered household locations. Shapes indicate whether at least one LASV-seropositive individual was recorded in the household. The color classification reflects the local clustering of seropositivity, where the Gi\* test includes the household's own value in the local average; a high-value central household may thus be classified as 'Not significant' if surrounding seronegative households dilute its local average below the statistical threshold. Scale bars are included to indicate spatial scale; coordinates are not shown to preserve privacy.

age does not act as a consistent biological or behavioral risk factor in isolation, but rather acts as a proxy for the varying duration and timing of exposure events at the community level.

Consistent with this theme of local heterogeneity, our univariable analyses revealed no strong or consistent demographic, environmental, or occupational correlates of seropositivity. The lack of clear associations with commonly cited risks, such as farming or rodent consumption (despite reported consumption up to 91% in BN and CR), contrasts with some previous findings and suggests that transmission may occur more frequently through indirect routes, such as food contamination [43]. However, the absence of associations may be a function of the study's limited statistical power to detect modest individual-level effects, given the low overall seroprevalence (N = 61) and limited evidence for prior effect sizes to guide adequate *a priori* powering. Furthermore, the near-universal reporting of certain exposures, such as the presence of rodents in homes (94%), creates little variability for comparison, mathematically obscuring potential effects. We must also acknowledge alternative explanations, specifically that self-reported behaviors are subject to recall bias and measurement error, and cross-sectional questionnaires may temporally misalign current habits with historical exposure events. Collectively, these findings suggest that if human behavior cannot be reliably used to predict risk in this setting, exposure is likely influenced by the primary missing component of the risk pathway: the local density and viral shedding status of the rodent reservoir [4,11].

Our findings highlight a vulnerability in the planning of Phase III vaccine efficacy trials. We found that being located within a high-risk state or Local Government Area does not guarantee high local exposure. We found that being located within a high-risk state or a historically highly-notified Local Government Area (such as Izzi LGA, which has reported confirmed cases in 7 different years) does not guarantee high, uniform local exposure across its constituent communities. Consequently, trial sites selected based on regional notification rates may inadvertently target villages with insufficient transmission intensity to demonstrate vaccine efficacy. To mitigate this risk, clinical trial site selection must adopt a One Health trial design framework [16]. Operationally, this shifts trial planning from a reactive reliance on lagging human clinical indicators to a predictive model driven by leading ecological and socio-behavioral data. A One Health approach integrates proactive ecological surveillance of the reservoir (e.g., tracking *M. natalensis* population dynamics and epizootic cycles), viral genomics, and continuous socio-behavioral monitoring to capture fluctuations in the human-animal interface.

## Strengths and limitations

The strengths of this study include its systematic, multi-criteria site selection; a large sample size across diverse communities; and a robust analytical framework using Bayesian mixed models to account for data hierarchies and uncertainty. The inclusion of formal spatial statistics adds rigor to our conclusion that transmission is not strongly clustered at the household level; however, we note that our ability to detect micro-clustering is inherently constrained by the low overall prevalence, the specific spatial scale of the analysis, and our quota sampling design, which captured only a subset of individuals per household. Nevertheless, the study has important limitations. It was powered to detect village-level differences based on higher seroprevalence estimates from previous studies; however, the observed low seroprevalence limited our power for individual-level risk factor analysis.

While our analysis was causally structured using *a priori* directed acyclic graphs [18], the cross-sectional design fundamentally limits precise temporal resolution and the distinction between recent and past infections. Consequently, while baseline seroprevalence is a superior screening tool compared to regional risk maps, it remains a proxy for true incidence. Ideally, serosurveys should be used to stratify villages for subsequent prospective incidence studies. This challenge is compounded by the potential for antibody waning and sero-reversion, the rates of which are vital for accurate FOI estimation but remain poorly understood for LASV. Additionally, the measurement of behavior and environment at a single cross-sectional point may not accurately reflect cumulative lifetime exposure, as housing and habits vary over time. Reliance on self-reported behaviors is subject to recall bias, and the study lacked objective ecological measures, such as rodent abundance data.

A further limitation is the potential for reduced diagnostic sensitivity associated with the use of DBS eluates. While DBS offers significant logistical advantages, elution from filter paper has been shown to result in a 4.6-fold reduction in the mean ELISA index value compared to matched whole-blood samples [26]. In head-to-head comparisons, this resulted in a lower proportion of positive detections (15% for DBS vs 29% for whole blood). Consequently, our reported seroprevalence of 3.3% should be viewed as a conservative estimate that may under-represent individuals with low antibody titers, potentially associated with more historical exposures. However, the high specificity of the ELISA (100% in reference cohorts) and high agreement with other platforms like Indirect Fluorescent Antibody (IFA) tests (93.1% overall agreement; $\kappa = 0.81$) ensures that the identified spatial hotspots and village-level differences are robust [26]. This high specificity is particularly important for vaccine trial site selection, where the risk of false-positive results from cross-reactive arenaviruses must be minimised.

Finally, the reduced sensitivity of DBS eluates has specific implications for the interpretation of age-stratified seroprevalence. If historical, low-titer antibodies are more likely to fall below the limit of detection, our findings may be subject to a recency bias. This could artificially exaggerate the appearance of peaks in younger cohorts while masking cumulative exposure in older individuals. When coupled with the estimated annual seroreversion rate of 3% [42], this technical limitation reinforces the view that our results represent a moving window of recent exposure rather than a complete record of lifetime risk. Consequently, while the divergent age patterns we observed strongly suggest local heterogeneity in transmission history, the absolute magnitude of cumulative exposure in older adults is likely under-represented.

### Public health implications and future directions

The absence of spatial clustering and a clear risk profile suggests that geographically-targeted human interventions may be inefficient. Instead, public health efforts should prioritize broadly-deployed, household-level interventions (e.g., rodent-proofing) and maintain high suspicion in surveillance across all villages within endemic regions, not just sentinel sites. Surveillance must also include areas of previously low-predicted risk given the observed micro-heterogeneity.

Our findings highlight a vulnerability in Phase III vaccine efficacy trials: being located within a high-risk state or LGA does not guarantee high local exposure. Trial sites selected based on regional notification rates may inadvertently target villages with insufficient transmission intensity to demonstrate efficacy. To mitigate this risk, clinical trial site selection must adopt the One Health trial design framework discussed above. By moving beyond passive surveillance data to incorporate granular, active human serosurveys alongside prospective, longitudinal monitoring of rodent reservoirs, researchers can ensure trials are enrolled in locations where the virus is demonstrably circulating in the animal host. This integrated strategy maximizes the likelihood of capturing spillover events and identifying active transmission hotspots. Future research must leverage data from ongoing longitudinal rodent sampling to link ecological fluctuations with human seroconversion [18].

### Conclusion

LASV exposure in rural Nigeria is highly variable and likely driven by complex, hyper-local factors. These findings caution against reliance on broad-scale risk mapping for public health planning. Effective control requires shifting toward adaptable, household-focused interventions and, critically, the adoption of granular, One Health surveillance strategies to accurately identify high-transmission sites for clinical research and vaccine deployment.

### Supporting information

**S1 Fig. Modelled village population sizes.** Estimates of the total population size for the nine surveyed villages, utilized to inform sampling frames and demographic quotas.
(JPG)

**S2 Fig. Participant flow diagram.** Flowchart detailing the systematic enrollment of households and individuals, including exclusions and the final number of participants included in the serological and risk factor analyses.
(JPG)

**S1 Table. Univariable analysis of risk factors for Lassa virus seropositivity.** Unadjusted Odds Ratios (OR) and 95% Credible Intervals (CrI) from Bayesian hierarchical models assessing demographic, environmental, occupational, and behavioral predictors of exposure.
(DOCX)

**S2 Table. Posterior summaries for prevalence models.** Median estimates and 95% Credible Intervals (CrI) on the probability scale (expressed as percentages) for overall, state-level, and village-level Lassa virus IgG seroprevalence.
(DOCX)

**S3 Table. Posterior summaries for age-seroprevalence GAM parameters.** Central estimates and 95% Credible Intervals (CrI) on the logit scale for the village-level intercepts and smooth term standard deviations extracted from the Bayesian Generalized Additive Model.
(DOCX)

**S4 Table. Spatial heterogeneity of risk factors between villages.** Posterior summaries of the random effect standard deviations (SD) quantifying the magnitude of between-village variance for the assessed explanatory variables.
(DOCX)

## Acknowledgments

We sincerely thank the residents and leadership of the nine study communities in Benue, Cross River, and Ebonyi states for their warm welcome, hospitality, and willingness to participate in this research. We are deeply grateful to the community liaisons and village guides whose local knowledge and dedication were vital for facilitating community entry and navigation. We also acknowledge the State Ministries of Health and the local public health teams, including the Disease Surveillance and Notification Officers (DSNOs), for their essential logistical support and ongoing collaboration throughout the fieldwork.

## Author contributions

**Conceptualization:** David Simons, Christina Harden, Natalie Imirzian, Lina Moses, David W. Redding, Sagan Friant.

**Data curation:** David Simons, Diana Marcus, Fisayomi Aderibigbe.

**Formal analysis:** David Simons.

**Funding acquisition:** Martin Meremikwu, Lina Moses, David W. Redding, Sagan Friant.

**Investigation:** David Simons, Christina Harden, Katharine E. T. Thompson, Nzube Michael Ifebueme, Sunday Eziechina, Helen Ignatius, Diana Marcus, Fisayomi Aderibigbe, James T. Koninga.

**Methodology:** David Simons, Christina Harden, Katharine E. T. Thompson, Sunday Eziechina, Sagan Friant.

**Project administration:** David Simons, Christina Harden, Natalie Imirzian, Nzube Michael Ifebueme, Sunday Eziechina, Martin Meremikwu, Sagan Friant.

**Supervision:** Martin Meremikwu, Lina Moses, David W. Redding, Sagan Friant.

**Visualization:** David Simons.

**Writing – original draft:** David Simons.

**Writing – review & editing:** Christina Harden, Natalie Imirzian, Katharine E. T. Thompson, Nzube Michael Ifebueme, Sunday Eziechina, Helen Ignatius, Diana Marcus, Fisayomi Aderibigbe, Martin Meremikwu, Lina Moses, David W. Redding, Sagan Friant.

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
