## [Decision Letter · Decision Letter 0]

8 Mar 2026

PNTD-D-26-00136

Local heterogeneity in Lassa fever serology in rural Nigeria: Implications for control strategies and vaccine trial site selection

Dear Dr. Simons,

Thank you for submitting your manuscript to PLOS Neglected Tropical Diseases. After careful consideration, we feel that it has merit but does not fully meet PLOS Neglected Tropical Diseases's publication criteria as it currently stands. Therefore, we invite you to submit a revised version of the manuscript that addresses the points raised during the review process.

Please submit your revised manuscript within by May 07 2026 11:59PM. If you will need more time than this to complete your revisions, please reply to this message or contact the journal office at plosntds@plos.org. Please include the following items when submitting your revised manuscript:

We look forward to receiving your revised manuscript.

Kind regards,

David Safronetz, Ph.D.

Section Editor

David Safronetz

Section Editor

Shaden Kamhawi

co-Editor-in-Chief

Paul Brindley

co-Editor-in-Chief

**Journal Requirements:**

1) Thank you for including an Ethics Statement for your study. Please include:

i) A statement that formal consent was obtained (must state whether verbal/written) OR the reason consent was not obtained (e.g. anonymity). NOTE: If child participants, the statement must declare that formal consent was obtained from the parent/guardian.].

4) Please amend your detailed Financial Disclosure statement. This is published with the article. It must therefore be completed in full sentences and contain the exact wording you wish to be published.

**Reviewers' Comments:**

Reviewer's Responses to Questions

**Key Review Criteria Required for Acceptance?**

**Methods**

-Are the objectives of the study clearly articulated with a clear testable hypothesis stated?

-Is the study design appropriate to address the stated objectives?

-Is the population clearly described and appropriate for the hypothesis being tested?

-Is the sample size sufficient to ensure adequate power to address the hypothesis being tested?

-Were correct statistical analysis used to support conclusions?

-Are there concerns about ethical or regulatory requirements being met?

Reviewer #1: The study was designed based on predicted models and estimates of seroprevalence to guide the number of participants needed for statistical power. However, the actual measured rates of seropositivity in individuals were much lower than expected, which made it difficult for the authors to derive any meaningful conclusions from the survey results.

Were correct statistical analysis used to support conclusions?

If the goal is to develop a model that helps to find which communities are expected to have higher exposure rates to LASV then interpreting the significance of individual predictors is not the right approach. The effect of predictors can be masked by other correlated predictors. Testing different multivariate models using an approach like cross-validation would be more useful as it would help to determine the out-of-sample accuracy.

Using significance tests (albeit Bayesian ones) as a basis to decide whether parameters are useful in a predictive model also ignores an advantage of regularizing priors, namely that predictors with low effects or weak evidence will have their coefficients shrunk toward 0 and little impact on the outcome. In this context, it might make more sense to use a full multivariate model as a baseline prediction model (including all the predictors the authors think are relevant) and evaluate which predictors can be removed without losing predictive power.

Reviewer #2: The manuscript by Simons and colleagues presents a cross-sectional serosurvey aimed at estimating the seroprevalence of Lassa fever virus (LASV) antibodies in rural Nigerian populations. This study is important for understanding the distribution of LASV in Nigeria, specifically within rural communities across the states of Benue, Ebonyi, and Cross River. Testing a total of 1,874 individuals, the study reports an overall seroprevalence of 3.3%, with substantial heterogeneity observed across both states and neighboring villages. While the findings are valuable, several issues require further clarification, which could strengthen the conclusions of the study.

The manuscript does not provide sufficient detail on the sensitivity and specificity of the commercial ELISA kit used for detecting LASV IgG antibodies. Given the potential impact of assay performance on the accuracy of seroprevalence estimates, it is crucial to address whether the assay was validated against samples from Lassa fever survivors or other well-established reference populations. The statement that "the high specificity of the commercial ELISA kit (LASV vs. non-LASV arenaviruses) may yield more conservative estimates than alternative serological assays" is problematic. The authors must provide actual sensitivity and specificity data to substantiate these claims. Without this information, it is difficult to assess whether the assay’s performance is adequate for the conclusions drawn, and whether the observed seroprevalence is reflective of true exposure rather than assay limitations.

Another critical issue is the use of dried blood spots (DBS) as a sampling method. The manuscript does not provide adequate detail on the validation of DBS for the detection of LASV IgG antibodies. Specifically, it is unclear whether the sensitivity and specificity of the ELISA assay were evaluated when using DBS samples, and how these may differ from serum samples. DBS is commonly used in field studies, but it is important to confirm that the performance of the assay with DBS is comparable to that with traditional blood collection methods. Given that the assay detected few positive cases, it is essential to determine whether the use of DBS could have contributed to this low detection rate.

Reviewer #3: The methods are generally well-described. The primary concern involves the use of dried blood spots (DBS) instead of serum. The authors should detail any modifications to the ELISA protocol in the methods section and provide the supporting validation data.

Reviewer #4: 1) The methods and overall study design and objective are clear and well presented.

2) The authors employ are relatively new commercial LASV IgG assay - that while potentially useful, is not well validated in terms of comparison to other assays. The authors should bring out this source of potential variation in light of differences with other studies completed using similar epidemiologic study designs in Nigeria.

3) Why were no attempts made (or at least not mentioned in this manuscript regarding IgM assessment?

3) Sample size and study population descriptions appear adequate and sufficient to support the study objectives.

**Results**

-Does the analysis presented match the analysis plan?

-Are the results clearly and completely presented?

-Are the figures (Tables, Images) of sufficient quality for clarity?

Reviewer #1: Why do the authors set "past_rodent_consumption" to missing if the individual reports current consumption of rodents? (file 07_descriptive_table_analysis.R lines 219-220) This seems like it dramatically reduces the number of cases (and therefore the power) for this predictor? It could be argued that, when looking at serology, past consumption is more relevant than current consumption of rodents since antibodies are accumulated over years and decades.

The authors do a good job on making the raw data and all analysis code public. However, the paper should provide citations and version numbers for the packages used, especially tidyverse, brms, RStan (used by brms). Many packages are maintained by funded projects and rely on the citation counts to justify their continued funding.

In Figure 1, the colours used make it difficult to identify the corresponding village lines and associated ribbon lines. The ribbon lines are hard to distinguish for Benue and Cross River especially. Please clarify or use different colours if possible.

For Figure 3, it is unclear what the different shapes and colours mean on each village map. For example, what does a blue triangle represents, or a red triangle? Also, having two circles represent two different parameters such as either “Not significant” or “Seropositive” makes it difficult to interpret. Please clarify.

Reviewer #2: The study finds no consistent demographic, environmental, or behavioral risk factors associated with seropositivity, including rodent consumption. However, this conclusion may be premature. The study does not provide a detailed analysis of the variables considered and how they were incorporated into the statistical models. It would be helpful to see a more comprehensive examination of the potential confounders or biases, as the lack of clear associations may be due to insufficient stratification or power. Further exploration of how these variables were defined and handled in the Bayesian hierarchical models would improve the interpretation of the results.

The manuscript lacks detailed information about the populations tested, especially considering Nigeria's detailed record-keeping on Lassa fever cases. The authors should provide additional context regarding the prevalence of Lassa fever in the surveyed regions and whether the seroprevalence aligns with expectations for these areas. It would also be beneficial to provide more context on whether the high or low prevalence states were overrepresented in the survey, and whether this could have influenced the findings. Greater transparency regarding the selection of villages and the demographics of the study population would help readers understand the external validity of the results.

Reviewer #3: All results are presented clearly and comprehensively.

Reviewer #4: 1) Overall the results present match the intended analysis plan and seem to provide adequate information to the reader to draw conclusions in a clear and concise manner.

2) Tables and figures appear appropriate for this type of epidemiologic serosurvey study.

**Conclusions**

-Are the conclusions supported by the data presented?

-Are the limitations of analysis clearly described?

-Do the authors discuss how these data can be helpful to advance our understanding of the topic under study?

-Is public health relevance addressed?

Reviewer #1: Some of the conclusions the authors draw about the shape of the estimated age seroprevalence curves appear somewhat unjustified. The only curve that appears not to include both positive and negative slopes is the one for Ogamanna. This could be due to an increasing chance of having been exposed as age increases (as the authors suggest but is absent from all other locations) or evidence of a wave of LASV exposures a few decades back. It is unclear why the authors suggest that a new resurgence of LASV in local rodents would lead to an inverted curve with young people more susceptible to exposure. One would suspect that if there is active, ongoing viral circulation within an area that all populations would be equally as susceptible? The authors do elude to the possibility that older individuals may be more “protected” due to historical exposures, however, there was no meaningful association found for age as a correlate of LASV seropositivity based on the data presented.

The authors do a good job of highlighting the limitations of the analysis – including acknowledging the lower than expected seropositivity rates which in turn limited the authors’ power for individual-level risk factor analysis.

The authors’ data shows that being located within a high-risk state or Local Government Area does not guarantee high local exposure, and may impact the utility of choosing these areas for potential clinical trial sites. The authors highlight the need for active serosurveys and longitudinal monitoring of rodent reservoirs to help site selection, rather than relying on passive surveillance data.

Reviewer #2: The paper makes reference to prior serosurveys but provides little in the way of direct comparative data. A more robust discussion of how these findings compare with previous studies would be valuable for placing the results in a broader context. It would be useful to include data from prior surveys that have tested similar populations and used similar methods, especially if they report contrasting findings. This comparison could also help clarify the heterogeneity in seroprevalence between villages and states.

Reviewer #3: The conclusions are well-supported by the data, and their public health implications are thoroughly discussed.

If the impact of DBS on assay sensitivity and specificity remains unquantified, this uncertainty should be explicitly addressed as a study limitation.

Reviewer #4: 1) Overall the conclusions on this article are insightful into the fine-scale variability of LASV exposure in the human population in the study area of Nigeria.

2) In this reviewers opinion - a major limitation that should be more explored or at least acknowledged more fully is the utilization of relatively new diagnostic assay modality for LASV IgG assessment. It would be worthwhile to provide a better comparison of this approach to previous serological tests to provide context for readers who are not likely to be familiar with these technical differences.

3) The authors overstate the "extreme" nature of the variability of seropositivity among the population. This reviewer appreciates that they are trying to draw attention to the fact that zoonotic virus spillover is indeed a stocastic process that lies on a gradient of risk. However - in each household examined typically only one individual was found to be seropositive and ratios of pos/neg individuals while variable were not "extreme". This type of verbiage is unnecessary and is overselling what the data supports.

4) The conclusions are informative about the challenges of placing upcoming LASV vaccine candidate Phase III efficacy trials at study sites. This is perhaps the most important conclusion from this work of relevance to public health. In the introduction the authors state that these findings influence trial "recruitment", but this is not correct. The key conclusion is that "site selection" will be critical to assess what locations may provide sufficient detections of acute LASV exposure and infection. Recruitment is simply the enrollment of the participants based on inclusion and exclusion critieria. Please adjust this text for clarity for the reader.

5) Much more informative and relevant to the current status of LASV force of infection in the communities studies would be to assess the detection of IgM in these communities. The authors should explicitly state why this was not done in the discussion and what that type of longitudinal followup could provide for trial study site selection etc. This is especially important considering that likely most LASV infections are mild to sub-clinical.

**Editorial and Data Presentation Modifications?**

Reviewer #1: In the abstract, under the Methods section, lines 26 to 33 is essentially repeated in the Results section, refer to lines 34 to 40. These sentences belong more to the results section and should be removed from the methods section.

Reviewer #2: The abstract contains several redundancies that detract from its clarity and conciseness. It would benefit from rewriting to eliminate repetitive statements and streamline the presentation of key findings. A clearer and more concise abstract would better highlight the manuscript's key contributions and help readers quickly understand the main conclusions.

Reviewer #3: (No Response)

Reviewer #4: 1) Overall, the authors tend to present their finding of local variation in LASV IgG as something that is somewhat surprising. This reviewer thinks that it would be helpful to moderate that overall tone a bit and put their findings in the context of what is seen with several other VHFs and other zoonotic pathogens. Clearly this is a critically important finding and will inform efforts to conduct rigorous site-selection for upcoming Phase III LASV vaccine candidate clinical trials. Adding some context would be helpful to strengthen the central take home message from this manuscript.

2) Supplementary Table 1: Why are data fields marked as "strong evidence" if the 95% CI include 1.0? Is this a data table presentation issue - or is the description on the methods sections regarding this table inaccurate (lines 192-194).

3) The authors mention several times about the need for "One Health" approaches to the implementation of human vaccine trial designs. The manuscript would be improved if a short description of what the authors mean by that statement could be included for clarity for readers who are not familiar with that subject.

**Summary and General Comments**

Reviewer #1: The goal of this research was to help identify potential sites for Phase III clinical trials to test LASV vaccines and to study potential local drivers of transmission. The authors wanted to assess if current broad-scale risk maps of LASV transmission can serve as good predictors for clinical trial site selection. In this study, a cross-sectional serosurvey was conducted including over 1800 indiviuals across 9 rural villages in Nigeria. These individuals were tested for levels of LASV IgG as a measure of past LASV infection and also participated in a questionnaire that covered various demographic, environmental, behavioural and occupational risk factors. The overall seropositivity rates were much lower than expected and extremely variable, as a result, no meaningful associations with risk factors included in the questionnaire were found.

The findings from this work is somewhat useful for those wishing to identify good sites for Phase III clinical trials for LASV vaccines, highlighting the deficiencies of relying on broad-based risk maps in Nigeria and the need for active surveillance not only in humans but in the rodent hosts as well. While insight is provided for vaccine trial site selection, limited control strategies are provided as implied in the title of this manuscript. Local causes of Lassa virus transmission remains poorly understood, although the authors acknowledge that other factors such as density of local rodent populations may be important but were not included in this study. Broad regional risk mapping or generalized risk profiles may not be representative of current transmission dynamics.

Reviewer #2: The lack of detailed assay validation, the unexplained use of dried blood spots, and the need for further exploration of the risk factors associated with seropositivity are all significant limitations. Additionally, further information about the population tested and comparison with prior serosurveys would strengthen the manuscript.

Reviewer #3: Simons et al. present a Lassa virus serosurvey across three Nigerian states, reporting an unexpectedly low seroprevalence. Although various factors were assessed for their association with Lassa virus exposure, no consistent patterns emerged. Overall, the study design and data analysis adhere to high standards, and the manuscript is well-written.

Minor point: Reference 13 is incomplete.

Reviewer #4: Overall, this manuscript describes the heterogeneity of accumulated life-time exposure to LASV as measured by IgG antibodies. This is an important finding relating to the eventual implementation of Phase III LASV vaccine candidate Phase III trials. Insights into the drivers of this finding were somewhat limited by poor correlation with specific risk factors, but perhaps not surprising given the overall very high rodent exposures across the study population over long time periods. A major improvement would be to add IgM detection among the study cohort to ascertain the levels of asymptomatic/sub-clinical infections in the study population.

PLOS authors have the option to publish the peer review history of their article (what does this mean?). If published, this will include your full peer review and any attached files.

**Do you want your identity to be public for this peer review?** For information about this choice, including consent withdrawal, please see our Privacy Policy.

Reviewer #1: No

Reviewer #2: No

Reviewer #3: No

Reviewer #4: No

**Figure resubmission:**
---

## [Decision Letter · Decision Letter 1]

10 May 2026

PNTD-D-26-00136R1

Local heterogeneity in Lassa fever serology in rural Nigeria: Implications for vaccine trial site selection

Dear Dr. Simons,

Thank you for submitting your manuscript to PLOS Neglected Tropical Diseases. After careful consideration, we feel that it has merit but does not fully meet PLOS Neglected Tropical Diseases's publication criteria as it currently stands. Therefore, we invite you to submit a revised version of the manuscript that addresses the points raised during the review process.

Please submit your revised manuscript within by Jun 09 2026 11:59PM. If you will need more time than this to complete your revisions, please reply to this message or contact the journal office at plosntds@plos.org. Please include the following items when submitting your revised manuscript:

As the corresponding author, your ORCID iD is verified in the submission system and will appear in the published article. PLOS supports the use of ORCID, and we encourage all coauthors to register for an ORCID iD and use it as well. Please encourage your coauthors to verify their ORCID iD within the submission system before final acceptance, as unverified ORCID iDs will not appear in the published article. Only the individual author can complete the verification step; PLOS staff cannot verify ORCID iDs on behalf of authors.

We look forward to receiving your revised manuscript.

Kind regards,

David Safronetz, Ph.D.

Section Editor

David Safronetz

Section Editor

Shaden Kamhawi

co-Editor-in-Chief

Paul Brindley

co-Editor-in-Chief

**Additional Editor Comments :**

Please review comments by reviewers 2 and address appropriately.

**Reviewers' Comments:**

Reviewer's Responses to Questions

**Key Review Criteria Required for Acceptance?**

**Methods**

-Are the objectives of the study clearly articulated with a clear testable hypothesis stated?

-Is the study design appropriate to address the stated objectives?

-Is the population clearly described and appropriate for the hypothesis being tested?

-Is the sample size sufficient to ensure adequate power to address the hypothesis being tested?

-Were correct statistical analysis used to support conclusions?

-Are there concerns about ethical or regulatory requirements being met?

Reviewer #2: The methods are appropriate. The use of LASV IgG seropositivity as the primary outcome is a major strength, as it captures the large burden of subclinical infection and therefore provides a much more accurate picture of exposure than case-based data. The selection of a high-specificity nucleoprotein-based ELISA is appropriate. False positives would undermine site selection. The analysis is fundamentally constrained by low seroprevalence, with only 61 seropositive individuals, which substantially limits statistical power. This makes it difficult to detect modest but meaningful associations across the 21 risk factors and increases the likelihood of type II error. This should be discussed in more detail.

Reviewer #3: (No Response)

**Results**

-Does the analysis presented match the analysis plan?

-Are the results clearly and completely presented?

-Are the figures (Tables, Images) of sufficient quality for clarity?

Reviewer #2: The analysis matches the plan. The results clearly and completely presented. THe Figures and Tables are of good quality.

Reviewer #3: (No Response)

**Conclusions**

-Are the conclusions supported by the data presented?

-Are the limitations of analysis clearly described?

-Do the authors discuss how these data can be helpful to advance our understanding of the topic under study?

-Is public health relevance addressed?

Reviewer #2: The empirical observations are effectively integrated into a broader One Health framework, with implications for vaccine trial design. However, several of its central interpretations extend beyond what the data can firmly support.

Reviewer #3: (No Response)

**Editorial and Data Presentation Modifications?**

Reviewer #2: The manuscript is well written and presented.

Reviewer #3: (No Response)

**Summary and General Comments**

Reviewer #2: The inclusion of 21 pre-specified risk factors spanning demographic, environmental, behavioural, and occupational domains is appropriate to avoid post hoc inference. Importantly, the lack of consistent associations is an informative result that challenges prevailing assumptions about spillover risk. Similarly, the spatial analysis is methodologically rigorous, distinguishing between broad and fine-scale clustering. The finding that only a small fraction of seropositive households fall within identified hotspots provides useful, if limited, evidence against simple spatial aggregation.

Major concerns are as follows:

1. Tthe conclusion that there are no consistent individual-level risk factors is too definitive. A more accurate interpretation is that the study was underpowered to detect such associations under conditions of low prevalence and limited exposure variability. The discussion should be modifued.

2. The near ubiquity of certain exposures, such as rodent presence in households, which further obscures potential effects. This should be discussed.

3. The interpretation that risk is driven primarily by stochastic, localized ecological factors is plausible but not directly demonstrated by the data. The absence of detectable associations with measured human behaviors does not necessarily imply that ecology is the dominant driver; alternative explanations include measurement error in self-reported behaviors, misalignment between current behaviors and past exposure, and the absence of direct ecological measurements such as rodent density or infection prevalence. This conclusion should be modified.

4. The conclusion that transmission is not spatially clustered should be treated with caution, as the ability to detect clustering is limited under low-prevalence conditions, may be sensitive to the choice of spatial scale, and is constrained by the sampling design, which captures only a subset of individuals per household. This conclusion should be modified.

5. The cross-sectional design limits causal inference, as seropositivity reflects past exposure without precise temporal resolution. This complicates interpretation of both risk factors and age-specific patterns, which may reflect a mixture of recent transmission, historical exposure, and antibody waning. Although the authors appropriately avoid formal force-of-infection modeling, they nonetheless draw relatively strong conclusions from age-seroprevalence curves that are not fully identifiable given the available data and should be modified.

6. The reduced sensitivity of dried blood spot sampling likely biases detection toward recent or higher-titer infections and may underrepresent cumulative exposure in older individuals, potentially exaggerating apparent peaks in younger age groups. This should be discussed.

Reviewer #3: The authors have addressed my comments and have incorporated them sufficiently into the manuscript.

PLOS authors have the option to publish the peer review history of their article (what does this mean?). If published, this will include your full peer review and any attached files.

Reviewer #2: No

Reviewer #3: No

**Figure resubmission:**
---

## [Editor Report · Decision Letter 2]

14 May 2026

Dear Dr. Simons,

We are pleased to inform you that your manuscript 'Local heterogeneity in Lassa fever serology in rural Nigeria: Implications for vaccine trial site selection' has been provisionally accepted for publication in PLOS Neglected Tropical Diseases.

Best regards,

David Safronetz, Ph.D.

Section Editor

David Safronetz

Section Editor

Shaden Kamhawi

co-Editor-in-Chief

Paul Brindley

co-Editor-in-Chief

---

## [Editor Report · Acceptance letter]

Dear Dr. Simons,

We are delighted to inform you that your manuscript, "Local heterogeneity in Lassa fever serology in rural Nigeria: Implications for vaccine trial site selection," has been formally accepted for publication in PLOS Neglected Tropical Diseases.

Best regards,

Shaden Kamhawi

co-Editor-in-Chief

Paul Brindley

co-Editor-in-Chief
